# Pricing Path-Dependent Options under Stochastic Volatility via Mellin Transform

Jiling Cao, Xi Li and Wenjun Zhang *

Department of Mathematical Sciences, School of Engineering, Computer and Mathematical Sciences, Auckland University of Technology, Private Bag 92006, Auckland 1142, New Zealand; jiling.cao@aut.ac.nz (J.C.); xi.li@aut.ac.nz (X.L.)
* Correspondence: wenjun.zhang@aut.ac.nz

**Abstract:** In this paper, we derive closed-form formulas of first-order approximation for down-and-out barrier and floating strike lookback put option prices under a stochastic volatility model using an asymptotic approach. To find the explicit closed-form formulas for the zero-order term and the first-order correction term, we use Mellin transform. We also conduct a sensitivity analysis on these formulas, and compare the option prices calculated by them with those generated by Monte-Carlo simulation.

**Keywords:** asymptotic approximation; barrier; down-and-out; floating strike; lookback; Mellin transform; stochastic volatility

**MSC:** 91G20; 41A60; 44A99; 91G60

## 1. Introduction

A standard option gives its owner the right to buy (or sell) some underlying asset in the future for a fixed price. Call options confer the right to buy the asset, while put options confer the right to sell the asset. Path-dependent options represent extensions of this concept. For example, a lookback call option confers the right to buy an asset at its minimum price over some time period. A barrier option resembles a standard option except that the payoff also depends on whether or not the asset price crosses a certain barrier level during the option's life. Lookback and barrier options are two of the most popular types of path-dependent options

Following the lead set by Black and Scholes (1973) and assuming that the underlying asset price follows a geometric Brownian motion with constant volatility, Merton (1973) derived a closed-form pricing formula for down-and-out call options. Reiner and Rubinstein (1991) extended Merton's results to other types of barrier options. Goldman et al. (1979) and Conze and Vishwanathan (1991) provided closed-form pricing formulas for lookback options. For a good summary for research on path-dependent options under the Black–Scholes framework, refer to Clewlow et al. (1994). As we know, the assumption that an asset price process follows a geometric Brownian motion with constant volatility does not capture the empirical observations, due to the volatility smile effect. So, it is desirable to overcome this drawback. There are different ways of extending the Black–Scholes model to incorporate the "smile" feature: one way is to consider "local volatility", and the other is to consider "stochastic volatility".

One popular local volatility model is the constant elasticity of variance (CEV) model introduced by Cox (1975, 1996), where a closed-form pricing formula for European call options was presented. Davydov and Linetsky (2001) derived solutions for barrier and lookback option prices under the CEV process in closed form and demonstrated that barrier and lookback option prices and hedge ratios under the CEV process can deviate dramatically from the lognormal values. In Boyle and Tian (1999), the pricing of certain

path-dependent options was re-examined when the underlying asset follows the CEV diffusion process, by approximating the CEV process using a trinomial method.

Heston (1993) assumes that volatility reverts to a long-term mean at a specified rate. Bates (1996) builds upon the Heston model by introducing a jump component for asset prices, which is represented as a compound Poisson process with normally distributed jumps. In a further refinement of the Heston model, the jumps are characterized by infinite activity jumps generated by a tempered stable process, as demonstrated in Zaevski et al. (2014). Despite these advancements, the pricing challenges associated with path-dependent options in the context of stochastic volatility persist, as no analytical solutions are available for these models.

Chiarella et al. (2012) considered the problem of numerically evaluating barrier option prices when the underlying dynamics are driven by the Heston stochastic volatility model and developed a method of lines approach to evaluate the price as well as the delta and gamma of the option. Park and Kim (2013) investigated a semi-analytic pricing method for lookback options in a general stochastic volatility framework. The resultant formula is well connected to the Black–Scholes price that is the first term of a series expansion, which makes computing the option prices relatively efficient. Furthermore, a convergence condition for the expansion was provided with an error bound. Leung (2013) and Wirtu et al. (2017) derived an analytic pricing formula for floating strike lookback options under the Heston model by means of the homotopy analysis method. The price is given by an infinite series whose value can be determined once an initial term is given well.

In addition, Kato et al. (2013) derived a new semi-closed-form approximation formula for pricing an up-and-out barrier option under a certain type of stochastic volatility model, including an SABR model. In a more recent paper by Funahashi and Higuchi (2018), a unified approximation scheme was proposed for a single-barrier option under local volatility models, stochastic volatility models, and their combinations. The basic idea of their approximation is to mimic a target underlying an asset process using a polynomial of the Wiener process. They then translated the problem of solving the first hit probability of the asset price into the problem of solving that of a Wiener process whose distribution of the passage time is known. Finally, utilizing Girsanov's theorem and the reflection principle, they showed that single-barrier option prices can be approximated in a closed form.

The main contribution of this paper is to derive new closed-form approximation formulas for pricing down-and-out put barrier options and floating strike lookback put options under a certain type of stochastic volatility model, which is similar to the one in Cao et al. (2023); Kato et al. (2013); Kim et al. (2023). To achieve our goal, we apply the asymptotic approach discussed in Fouque et al. (2011) and Mellin transform. Mellin transform techniques were used by Panini and Srivastav (2004) to derive integral equation representations for the price of European and American basket put options. Similarly, Yoon (2014) applied Mellin transform to derive a closed-form solution of the option price with respect to a European call option and a European put option with the Hull–White stochastic interest rate. Moreover, Kim and Yoon (2018) derived a closed-form formula of a second-order approximation for a European corrected option price under a stochastic elasticity of variance (SEV) model.

The rest of the paper is organized as follows. Section 2 discusses the model framework and the features of down-and-out and floating strike lookback put options. In Section 3, we provide detailed discussions on an asymptotic approach, which is used to derive approximations to the risk-netural values of these types of options. In Section 4, we apply Mellin transform to derive a closed-form formula of the first-order approximation for down-and-out barrier put options. In Section 5, we apply Mellin transform to derive a closed-form formula of the first-order approximation for floating strike lookback put options. Section 6 presents a sensitivity and comparison analysis and demonstrates that the results given by these closed-form formulas match well with those generated by Monte-Carlo simulation. Section 7 gives a brief summary. Details on Mellin transform and the derivation of the closed-form formulas in Sections 4 and 5 are provided in Appendices A and B, respectively.

## 2. Basic Model Set-Up and Path-Dependent Options

### 2.1. Stochastic Volatility Model

Let $\{S_t : t \geq 0\}$ denote the price process of a risky asset on some filtered probability space $(\Omega, \mathscr{F}, (\mathscr{F}_t)_{t \geq 0}, \mathbb{P})$, where $\mathbb{P}$ is the physical probability measure In this paper, we assume that $\{S_t : t \geq 0\}$ evolves according to the following system of stochastic differential equations:

$$
\begin{aligned}
dS_t &= \mu S_t dt + f(Y_t) S_t dW_t^s, \\
dY_t &= \alpha(m - Y_t) dt + \beta\left(\rho dW_t^s + \sqrt{1 - \rho^2} dW_t^y\right),
\end{aligned}
\tag{1}
$$

where $\mu$, $\alpha > 0$, $\beta > 0$, and $m$ are constants and $f$ is a function having positive values and specifying the dependence on the hidden process $\{Y_t : t \geq 0\}$. The processes $\{W_t^s : t \geq 0\}$ and $\left\{W_t^y : t \geq 0\right\}$ are independent standard Brownian motions. The constant correlation coefficient $\rho$ with $-1 < \rho < 1$ captures the leverage effect. Here, $\mu$ is the drift rate. The mean-reversion process $\{Y_t : t \geq 0\}$ given in Equation (1) is characterized by its typical time to return back to the mean level $m$ of its long-run distribution. The parameter $\alpha$ determines the speed of mean-reversion, and $\beta$ controls the volatility of $\{Y_t : t \geq 0\}$. In the sequel, we shall refer to the above system as the stochastic volatility (SV) model. In Sections 2 and 3, we will not specify the concrete form of $f$, but assume that $f$ is bounded and smooth enough, e.g., $f \in C_0^2(\mathbb{R})$. Furthermore, $f$ has to satisfy a sufficient growth condition in order to avoid bad behavior such as the non-existence of moments of $\{S_t : t \geq 0\}$. For numerical results in Section 6, we choose $f$ to take a special form, as used in Fouque et al. (2000, 2011) and Cao et al. (2021).

We apply the well-known Girsanov theorem to change the physical measure $\mathbb{P}$ to a risk-neutral martingale measure $\mathbb{Q}$ by letting

$$
dW_t^{s*} = \frac{\mu - r}{f(Y_t)} dt + dW_t^s y \quad \text{and} \quad dW_t^{y*} = \xi(Y_t) dt + dW_t^y,
$$

where $\xi(Y_t)$ represents the premium of volatility risk. Then, the model equations under the measure $\mathbb{Q}$ can be written as

$$
\begin{aligned}
dS_t &= rS_t dt + f(Y_t) S_t dW_t^{s*}, \\
dY_t &= \left[\alpha(m - Y_t) - \beta\left(\rho \frac{\mu - r}{f(Y_t)} + \xi(Y_t)\sqrt{1 - \rho^2}\right)\right] dt \\
&\quad + \beta\left(\rho dW_t^{s*} + \sqrt{1 - \rho^2} dW_t^{y*}\right).
\end{aligned}
\tag{2}
$$

Note that $\{W_t^{s*} : t \geq 0\}$ and $\left\{W_t^{y*} : t \geq 0\right\}$ are independent standard Brownian motions under $\mathbb{Q}$. As an Ornstein–Uhlenbeck (OU) process, $\{Y_t : t \geq 0\}$ in Equation (1) has an invariant distribution, which is normal with mean $m$ and variance $\beta^2/2\alpha$. Thus, we can expect that if mean reversion is very fast, i.e., $\alpha$ goes to infinity, the process $\{S_t : t \geq 0\}$ should be close to a geometric Brownian motion. This means that if mean reversion is extremely fast, then the model of Black and Scholes would become a good approximation. In reality, however, it may not be the case. For fast but not extremely fast mean-reversion, the Black–Scholes model needs to be corrected to account for the random characteristics of the volatility of a risky asset. For this purpose, we introduce another small parameter $\epsilon$ defined by $\epsilon = 1/\alpha$, as performed by Fouque et al. (2000). For notational convenience, we put $\nu = \beta/\sqrt{2\alpha}$. With the help of these notations, the model equations under $\mathbb{Q}$ are re-written as

$$
\begin{aligned}
dS_t &= rS_t dt + f(Y_t) S_t dW_t^{s*}, \\
dY_t &= \left[ \frac{1}{\epsilon}(m - Y_t) - \frac{\sqrt{2}\nu}{\sqrt{\epsilon}} \Lambda(Y_t) \right] dt + \frac{\sqrt{2}\nu}{\sqrt{\epsilon}} \left( \rho dW_t^{s*} + \sqrt{1 - \rho^2} dW_t^{y*} \right),
\end{aligned}
$$

where $\Lambda(\cdot)$, defined by

$$
\Lambda(y) := \rho \frac{\mu - r}{f(y)} + \xi(y) \sqrt{1 - \rho^2},
$$

is the combined market price of risk.

### 2.2. Path-Dependent Options

Let $V(T)$ denote the payoff of a put option on the risky asset at its expiration $T$. Then, its risk-neutral price at time $t \in [0, T]$ under our SV model is given by

$$
P(t, s, y) = \mathbb{E}^{\mathbb{Q}} \left( e^{-r(T-t)} V(T) \mid S_t = s, Y_t = y \right).
$$

Note that $V(T)$ depends on the type of options. In this paper, we consider two types of path-dependent options: down-and-out put options and floating strike lookback put options. For notational convenience, we put $U_t := \min_{0 \leq u \leq t} S_u$ and $Z_t := \max_{0 \leq u \leq t} S_u$. The payoff of a down-and-out put option is given by

$$
DOP(T) := \max\{K - S_T, 0\} \times \mathbb{1}_{U_T > B},
$$

where $K$ is the strike price, $B$ is the barrier level satisfying $0 < B < K$, and $\mathbb{1}_{U_T > B}$ is the indicator function. For a floating strike lookback put option, its payoff has the form of $LP_{float}(T) := Z_T - S_T$. Applying Itô's lemma, we can obtain a partial differential equation (PDE) for $P(t, s, y)$ as follows:

$$
\begin{aligned}
0 &= \frac{\partial P}{\partial t} + \frac{1}{2} s^2 f^2(y) \frac{\partial^2 P}{\partial s^2} + r \left( s \frac{\partial P}{\partial s} - P \right) + \frac{\sqrt{2}\rho \nu s}{\sqrt{\epsilon}} f(y) \frac{\partial^2 P}{\partial s \partial y} \\
&\quad + \frac{\nu^2}{\epsilon} \frac{\partial^2 P}{\partial y^2} + \left( \frac{1}{\epsilon}(m - y) - \frac{\sqrt{2}\nu}{\sqrt{\epsilon}} \Lambda(y) \right) \frac{\partial P}{\partial y}.
\end{aligned} \tag{3}
$$

The boundary conditions for Equation (3) vary depending on the type of options. For example, the boundary conditions for Equation (3) when $V(T) = DOP(T)$ are

$$
\begin{cases}
P(T, s, y) = \max\{K - s, 0\}, & s > B, \\
P(t, B, y) = 0, & 0 \leq t \leq T.
\end{cases}
$$

When $V(T) = LP_{float}(T)$, the boundary conditions become the following:

$$
\begin{cases}
\dfrac{\partial P}{\partial z}(t, z, y, z) = 0, & 0 \leq t \leq T, z > 0, \\
P(T, s, y, z) = z - s, & 0 \leq s \leq z.
\end{cases}
$$

Note that in this case, $P$ is a function of $t$, $s$, $y$, and $z$ (here, $Z_t = z$).

Remark: Since the Mellin transform of the payoff function of a call option is not defined, this paper primarily concentrates on evaluating put options. However, as outlined in Buchen (2001), the pricing of call options can be directly derived from put options through the put–call parity relationship.

## 3. Asymptotic Expansions

In this section, we apply an asymptotic expansion approach to establish partial differential equations, which will be used to derive an approximate solution to Equation (3) and thus find an approximated value of a put option.

We begin with re-organizing Equation (3) in terms of the orders of $\epsilon$ as follows:

$$\frac{1}{\epsilon}\mathcal{L}_0 P + \frac{1}{\sqrt{\epsilon}}\mathcal{L}_1 P + \mathcal{L}_2 P = 0, \tag{4}$$

where the operators $\mathcal{L}_0$, $\mathcal{L}_1$ and $\mathcal{L}_2$ are defined by

$$\begin{aligned}
\mathcal{L}_0 &:= (m-y)\frac{\partial}{\partial y} + v^2\frac{\partial^2}{\partial y^2}, \\
\mathcal{L}_1 &:= \sqrt{2}\rho v s f(y)\frac{\partial^2}{\partial s \partial y} - \sqrt{2}v\Lambda(y)\frac{\partial}{\partial y}, \text{ and} \\
\mathcal{L}_2 &:= \frac{\partial}{\partial t} + \frac{1}{2}s^2 f^2(y)\frac{\partial^2}{\partial s^2} + r\left(s\frac{\partial}{\partial s} - \cdot\right).
\end{aligned}$$

In order to obtain an efficient approximate solution to $P$, as that in Fouque et al. (2011), we apply the following asymptotic expansion of $P$:

$$P = P_0 + \sqrt{\epsilon}P_1 + \epsilon P_2 + \epsilon\sqrt{\epsilon}P_3 + \dots, \tag{5}$$

where $P_0$, $P_1$, ... are functions corresponding to varying orders of $\epsilon$. Substituting $P$ in Equation (5) into Equation (4) and re-organizing the terms, we obtain

$$\begin{aligned}
0 = &\frac{1}{\epsilon}\mathcal{L}_0 P_0 + \frac{1}{\sqrt{\epsilon}}(\mathcal{L}_1 P_0 + \mathcal{L}_0 P_1) + (\mathcal{L}_0 P_2 + \mathcal{L}_1 P_1 + \mathcal{L}_2 P_0) \\
&+ \sqrt{\epsilon}(\mathcal{L}_0 P_3 + \mathcal{L}_1 P_2 + \mathcal{L}_2 P_1) + \dots.
\end{aligned} \tag{6}$$

Our aim is to find $P_0$ and $P_1$.

Firstly, from the $O(1/\epsilon)$-order term in Equation (6), we obtain $\mathcal{L}_0 P_0 = 0$. If we assume that $P_0$ does not grow as fast as $e^{y^2/2}$, as was assumed in Choi et al. (2013), we can show that $P_0$ is independent of $y$. Secondly, from the $O(1/\sqrt{\epsilon})$-order term in Equation (6), we obtain $\mathcal{L}_1 P_0 + \mathcal{L}_0 P_1 = 0$. Since $P_0$ is independent of $y$, then $\mathcal{L}_1 P_0 = 0$. It follows that $\mathcal{L}_0 P_1 = 0$. Again, if we assume that $P_1$ does not grow as fast as $e^{y^2/2}$, then we can deduce that $P_1$ is also independent of $y$.

Next, from the $O(1)$-order term in Equation (6), we obtain

$$\mathcal{L}_0 P_2 + \mathcal{L}_1 P_1 + \mathcal{L}_2 P_0 = 0.$$

Since $P_1$ is independent of $y$, we have $\mathcal{L}_1 P_1 = 0$, which implies that

$$\mathcal{L}_0 P_2 + \mathcal{L}_2 P_0 = 0. \tag{7}$$

Seeing Equation (7) as a Poisson equation for $P_2$ in $y$, in order for it to have a solution, it is required to satisfy the centering condition

$$\langle \mathcal{L}_2 P_0 \rangle = \langle \mathcal{L}_2 \rangle P_0 = 0, \tag{8}$$

which is equivalent to

$$\frac{\partial P_0}{\partial t} + rs\frac{\partial P_0}{\partial s} + \frac{1}{2}s^2\langle f^2 \rangle\frac{\partial^2 P_0}{\partial s^2} - rP_0 = 0. \tag{9}$$

This is an equation for us to determine the $P_0$ term. Here, $\langle \cdot \rangle$ denotes the expectation with respect to the invariant distribution of the process $\{Y_t : t \geq 0\}$, i.e.,

$$\langle h \rangle = \int_{-\infty}^{+\infty} h(y)\Phi(y)dy, \quad \text{where} \quad \Phi(y) = \frac{1}{\sqrt{2\pi v^2}}e^{-\frac{(y-m)^2}{2v^2}}.$$

Note that a small $\epsilon$ value corresponds to fast-mean reverting. In this case, $Y_t$ approaches a constant and $\langle f^2 \rangle$ can be regarded as constant variance, and then Equation (9)

is the Black–Scholes PDE. Thus, for small $\epsilon$, $P_0$ represents the put option price under the Black–Scholes model.

Following Equation (8), we have

$$\mathcal{L}_2 P_0 = \mathcal{L}_2 P_0 - \langle \mathcal{L}_2 \rangle P_0 = \frac{1}{2}\left(f^2 - \langle f^2 \rangle\right)s^2 \frac{\partial^2 P_0}{\partial s^2},$$

which, together with Equation (7), implies

$$\mathcal{L}_0 P_2 = -\frac{1}{2}\left(f^2 - \langle f^2 \rangle\right)s^2 \frac{\partial^2 P_0}{\partial s^2}. \tag{10}$$

The solution to Equation (10) can be expressed as

$$P_2 = -\frac{1}{2}(\phi + c)s^2 \frac{\partial^2 P_0}{\partial s^2}, \tag{11}$$

where $\phi$ is a function of $y$ which only satisfies the equation $\mathcal{L}_0 \phi = f^2 - \langle f^2 \rangle$, and $c$ is a function of other variables except $y$.

To derive an equation for $P_1$, we consider the $O(\sqrt{\epsilon})$-term in Equation (6) and obtain

$$\mathcal{L}_0 P_3 + \mathcal{L}_1 P_2 + \mathcal{L}_2 P_1 = 0.$$

This equation can be regarded as a Poisson equation for $P_3$ in $y$, and in order for it to have a solution, the following centering condition must be satisfied:

$$\langle \mathcal{L}_1 P_2 + \mathcal{L}_2 P_1 \rangle = 0. \tag{12}$$

After we substitute $P_2$ in Equation (11) into Equation (12) and make simplifications, we obtain

$$\frac{\partial P_1}{\partial t} + \frac{1}{2}\langle f^2 \rangle s^2 \frac{\partial^2 P_1}{\partial s^2} + rs\frac{\partial P_1}{\partial s} - rP_1 = c_1 s^3 \frac{\partial^3 P_0}{\partial s^3} + c_2 s^2 \frac{\partial^2 P_0}{\partial s^2}, \tag{13}$$

where

$$c_1 := \frac{\sqrt{2}}{2}\langle f\phi' \rangle \rho \nu \quad \text{and} \quad c_2 := \frac{\sqrt{2}}{2}\left(2\rho\langle f\phi' \rangle - \langle \Lambda \phi' \rangle\right)\nu. \tag{14}$$

This is an equation for us to determine the first correction term $P_1$.

We summarize the previous formal analysis as the following theorem.

**Theorem 1.** *Under the SV model governed by Equation* (1)*, an approximation of the risk-neutral value P of a path-dependent put option is given by*

$$P = P_0 + \sqrt{\epsilon}P_1 + o(\sqrt{\epsilon}), \tag{15}$$

*for small $\epsilon$, where $P_0$ and $P_1$ are determined by Equations* (9) *and* (13) *with corresponding boundary conditions, respectively, such that $P_0$ is the put option price under the Black–Scholes model with constant effective volatility $\sqrt{\langle f^2 \rangle}$ and $P_1$ is the first-order correction term.*

Finally, as mentioned in Section 2, boundary conditions for Equations (8) and (13) depend on the types of options that we consider. We describe the corresponding boundary conditions and solve these equations in the next two sections.

## 4. Determining $P_0$ and $P_1$ for Down-and-Out Put Options

In this section, we use Mellin transform to derive analytical expressions of the $P_0$ and $P_1$ terms for down-and-out put options

### 4.1. $P_0$ Term for Down-and-Out Put Options

In order to use Mellin transform to calculate the $P_0$ term for down-and-out put options, noting that $P_0$ is independent of $y$ under our assumption, we first follow the method in Buchen (2001) and use the boundary condition,

$$P(T,s,y) = \max\{K - s, 0\}, \quad \text{for} \quad s > B,$$

to set up the boundary condition of $P_0$ for $s \geq 0$ as follows:

$$P_0(T,s) := (K - s)\mathbb{1}_{B<s<K} - \left(\frac{B}{s}\right)^{k_1-1}\left(K - \frac{B^2}{s}\right)\mathbb{1}_{\frac{B^2}{K}<s<B}, \tag{16}$$

where $k_1 = 2r/\langle f^2 \rangle$. Now, we apply Mellin transform to Equation (9) to convert this PDE into the following ODE:

$$\frac{d\hat{P}_0}{dt} + \left(\frac{1}{2}\langle f^2 \rangle(w^2 + w) - rw - r\right)\hat{P}_0 = 0. \tag{17}$$

The solution to Equation (17) is given by

$$\hat{P}_0(t,w) = \hat{\theta}(w)e^{\frac{1}{2}\langle f^2 \rangle\left(w^2 + (1-k_1)w - k_1\right)(T-t)}, \tag{18}$$

where $\hat{\theta}$ is a function of $w$, determined by the boundary condition (16).

Next, we take inverse Mellin transform of Equation (18) and obtain

$$P_0(t,s) = P_0(T,s) * \mathcal{M}^{-1}e^{\lambda(w+\eta)^2 + \delta},$$

where

$$\lambda = \frac{1}{2}\langle f^2 \rangle(T - t), \; \eta = \frac{1 - k_1}{2}, \; \delta = -\lambda\eta^2 - r(T - t)$$

and the operation $*$ means the convolution. Applying Table A1 in Appendix A and the boundary condition given in Equation (16), we have

$$
\begin{aligned}
P_0(t,s) &= P_0(T,s) * \left(\frac{e^\delta s^\eta}{2\sqrt{\lambda\pi}}e^{-\frac{1}{4\lambda}(\ln s)^2}\right) \\
&= \int_B^K (K - u)e^\delta \left(\frac{s}{u}\right)^\eta \left(\frac{1}{2\sqrt{\lambda\pi}}e^{-\frac{1}{4\lambda}\left(\ln\left(\frac{s}{u}\right)\right)^2}\right)\frac{du}{u} - \\
&\quad \int_{\frac{B^2}{K}}^B \left(\frac{B}{u}\right)^{k_1-1}\left(K - \frac{B^2}{u}\right)e^\delta \left(\frac{s}{u}\right)^\eta \left(\frac{1}{2\sqrt{\lambda\pi}}e^{-\frac{1}{4\lambda}\left(\ln\left(\frac{s}{u}\right)\right)^2}\right)\frac{du}{u}.
\end{aligned} \tag{19}
$$

After some careful calculation, for down-and-out put options, we derive a closed-form expression of the $P_0$ term as follows:

$$
\begin{aligned}
P_0(t,s) &= Ke^{-r(T-t)}\left(\Phi\left(-\Delta_-\left(\frac{s}{K}\right)\right) - \Phi\left(-\Delta_-\left(\frac{s}{B}\right)\right)\right) - \\
&\quad s\left(\Phi\left(-\Delta_+\left(\frac{s}{K}\right)\right) - \Phi\left(-\Delta_+\left(\frac{s}{B}\right)\right)\right) - \\
&\quad Ke^{-r(T-t)}\left(\frac{B}{s}\right)^{k_1-1}\left[\Phi\left(\Delta_-\left(\frac{B}{s}\right)\right) - \Phi\left(\Delta_-\left(\frac{B^2}{sK}\right)\right)\right] + \\
&\quad B\left(\frac{B}{s}\right)^{k_1}\left[\Phi\left(\Delta_+\left(\frac{B}{s}\right)\right) - \Phi\left(\Delta_+\left(\frac{B^2}{sK}\right)\right)\right],
\end{aligned} \tag{20}
$$

where $\Phi(\cdot)$ is the CDF of the standard normal distribution and

$$\Delta_\pm(x) = \frac{1}{\sqrt{\langle f^2 \rangle(T - t)}}\left[\ln(x) + \left(r \pm \frac{1}{2}\langle f^2 \rangle\right)(T - t)\right].$$

Note that $P_0$ given in Equation (20) is precisely the same as the price of a down-and-out put option given in the literature, e.g., Hull (2015, chp. 26, p. 606) or Haug (2006, chp. 4), if we let $\sigma^2 = \langle f^2 \rangle$. For details of the derivation of formula (20), we refer the reader to Appendix B.

*4.2. $P_1$ Term for Down-and-Out Put Options*

For down-and-out put options, the boundary conditions for $P_1$ are

$$\begin{cases} P_1(T,s) &=& 0, \quad for \ s \geq B, \\ P_1(t,B) &=& 0, \quad for \ 0 < t < T. \end{cases}$$

We again follow the method in Buchen (2001) and extend the boundary conditions $P_1(T,s) = 0$, for $s \geq B$ as $P_1(T,s) = 0$ for all $s \geq 0$.

Next, we apply Mellin transform to Equation (13) to obtain

$$\frac{d\hat{P}_1}{dt} + \left( \frac{1}{2}\langle f^2 \rangle \left( w^2 + w \right) - rw - r \right)\hat{P}_1 = (-c_1 w(w+1)(w+2) + c_2 w(w+1))\hat{P}_0.$$

Solving this equation, we obtain

$$\hat{P}_1(t,w) = \left[ c_1(T-t)w^3 - (c_2 - 3c_1)(T-t)w^2 - (c_2 - 2c_1)(T-t)w \right]\hat{P}_0(t,w).$$

Finally, applying inverse Mellin transform, we obtain an explicit closed-form expression of $P_1$ as follows:

$$\begin{aligned} P_1(t,s) &=& \mathcal{M}^{-1}\left(\hat{P}_1(t,w)\right) \\ &=& c_1(T-t)\left(-s\frac{d}{ds}P_0(t,s) - 3s^2\frac{d^2}{ds^2}P_0(t,s) - s^3\frac{d^3}{ds^3}P_0(t,s)\right) \\ && -(c_2 - 3c_1)(T-t)\left(s\frac{d}{ds}P_0(t,s) + s^2\frac{d^2}{ds^2}P_0(t,s)\right) \\ && -(c_2 - 2c_1)(T-t)\left(-s\frac{d}{ds}P_0(t,s)\right), \end{aligned} \tag{21}$$

where $P_0$ is given in the previous section and $c_1$ and $c_2$ are given in Equation (14).

We summarize the above analysis and calculation on down-and-out put options in the following theorem.

**Theorem 2.** *Under the SV model governed by Equation (1), an approximation of the risk-neutral value P of a down-and-out barrier put option is given by*

$$P = P_0 + \sqrt{\epsilon}P_1 + o(\sqrt{\epsilon}), \tag{22}$$

*where $P_0$ and $P_1$ are given by Equations (20) and (21), respectively.*

## 5. Determining $P_0$ and $P_1$ for Lookback Put Options

In this section, we use Mellin transform to derive analytical expressions of the $P_0$ and $P_1$ terms for floating strike lookback put options

*5.1. $P_0$ Term for Lookback Put Options*

For lookback floating strike put options, the boundary conditions of $P_0$ are

$$\begin{cases} \dfrac{\partial P_0}{\partial z}(t,z,z) &=& 0, \\ \dfrac{\partial P_0}{\partial z}(T,s,z) &=& 1, \quad for \ 0 < s < z. \end{cases}$$

Similar to the case of down-and-out put options, we extend the second boundary condition to $0 < s < \infty$ as follows:

$$\frac{\partial P_0}{\partial z}(T,s,z) := \mathbb{1}_{s<z} - \left(\frac{z}{s}\right)^{k_1-1} \cdot \mathbb{1}_{z<s}, \quad \text{for} \quad 0 < s < \infty.$$

Then, by integrating each side of the last equation, we can obtain

$$P_0(T,s,z) = \int_s^z -\left(\frac{\xi}{s}\right)^{k_1-1} d\xi = -\frac{1}{k_1}\left(\frac{z}{s}\right)^{k_1} s + \frac{1}{k_1}s \tag{23}$$

for $s > z$. For convenience, we let $u = s/z$ and $Q_0 = P_0/z$. With these notations, Equation (9) becomes

$$\frac{\partial Q_0}{\partial t} + \frac{1}{2}u^2\langle f^2\rangle\frac{\partial^2 Q_0}{\partial u^2} + ru\frac{\partial Q_0}{\partial u} - rQ_0 = 0, \tag{24}$$

with boundary conditions

$$Q_0(T,u) = -\frac{1}{k_1}u^{1-k_1} + \frac{1}{k_1}u, \quad \text{for} \quad u > 1, \tag{25}$$

and $Q_0(T,u) = 1$, for $0 < u < 1$.

Note that except the boundary conditions, Equation (24) is identical to Equation (9). Applying Mellin transform in the same way as that for the case of down-and-out put options, we can derive the solution to Equation (24) as follows:

$$Q_0(t,u) = \hat{\theta}(w) * \mathcal{M}^{-1} e^{\lambda(w+\eta)^2+\delta}.$$

Again, applying Table A1 and $P_0$ given in Equation (16), we have

$$\begin{aligned} Q_0(t,u) &= Q_0(T,u) * e^{\delta}z^{\eta}\left(\frac{1}{2\sqrt{\pi}}\lambda^{-\frac{1}{2}}e^{-\frac{1}{4\lambda}(\ln z)^2}\right) \\ &= \int_0^1 (1-\xi)e^{\delta}\left(\frac{u}{\xi}\right)^{\eta}\left(\frac{1}{2\sqrt{\pi}}\lambda^{-\frac{1}{2}}e^{-\frac{1}{4\lambda}\left(\ln\left(\frac{u}{\xi}\right)\right)^2}\right)\frac{d\xi}{\xi} + \\ &\quad \int_1^{\infty}\left(\frac{-1}{k_1}\xi^{1-k_1} + \frac{\xi}{k_1}\right)e^{\delta}\left(\frac{u}{\xi}\right)^{\eta}\left(\frac{1}{2\sqrt{\pi}}\lambda^{-\frac{1}{2}}e^{-\frac{1}{4\lambda}\left(\ln\left(\frac{u}{\xi}\right)\right)^2}\right)\frac{d\xi}{\xi}. \end{aligned} \tag{26}$$

After calculating integrals, for floating strike lookback put options, we derive a closed-form expression of the $P_0$ term as follows:

$$\begin{aligned} P_0(t,s,z) &= ze^{-r(T-t)}\Phi\left(-\Delta_-\left(\frac{s}{z}\right)\right) - s\Phi\left(-\Delta_+\left(\frac{s}{z}\right)\right) \\ &\quad -\frac{z}{k_1}\left(\frac{s}{z}\right)^{1-k_1}e^{-r(T-t)}\Phi\left(-\Delta_-\left(\frac{z}{s}\right)\right) + \frac{s}{k_1}\Phi\left(\Delta_+\left(\frac{s}{z}\right)\right), \end{aligned} \tag{27}$$

where $\Phi(\cdot)$ is the CDF of the standard normal distribution. Note that $P_0$ given in Equation (27) is precisely the same as the price of a floating strike put option given in the literature, e.g., Hull (2015, chp. 26, p. 608) or Haug (2006, chp. 4), if we let $\sigma^2 := \langle f^2\rangle$. Details of the derivation of this formula can be found in Appendix B.

### 5.2. $P_1$ Term for Lookback Put Options

For floating strike lookback put options, the boundary conditions for $P_1$ are

$$\begin{cases} P_1(T,s,z) &= 0, \quad \text{for } 0 < s < z, \\ \frac{\partial P_1}{\partial z}(t,z,z) &= 0, \quad \text{for } 0 < t < T \text{ and } z > 0. \end{cases}$$

Just like that for the $P_0$-term for floating strike lookback put options, we let $u = s/z$ and $Q_1 = P_1/z$. With these notation changes, Equation (13) is converted to the following:

$$\frac{\partial Q_1}{\partial t} + \frac{1}{2}\langle f^2 \rangle u^2 \frac{\partial^2 Q_1}{\partial u^2} + ru\frac{\partial Q_1}{\partial u} - rQ_1 = c_1 u^3 \frac{\partial^3 Q_0}{\partial u^3} + c_2 u^2 \frac{\partial^2 Q_0}{\partial u^2} \tag{28}$$

with $Q_1(T, u) = 0$ for $0 < u < 1$.

Note that Equation (28) is essentially the same as Equation (13), except the notational difference. So, we have

$$
\begin{aligned}
Q_1(t, u) \;=\; & c_1(T - t)\left(-u\frac{d}{du}Q_0(t, u) - 3u^2\frac{d^2}{du^2}Q_0(t, u) - u^3\frac{d^3}{du^3}Q_0(t, u)\right) \\
& -(c_2 - 3c_1)(T - t)\left(u\frac{d}{du}Q_0(t, u) + u^2\frac{d^2}{dz^2}Q_0(t, u)\right) \\
& -(c_2 - 2c_1)(T - t)\left(-u\frac{d}{du}Q_0(t, u)\right),
\end{aligned}
\tag{29}
$$

where $Q_0$ is given previously. Consequently, we have

$$
\begin{aligned}
P_1(t, s, z) \;=\; & c_1(T - t)\left(-s\frac{d}{ds}P_0(t, s, z) - 3s^2\frac{d^2}{ds^2}P_0(t, s, z) - s^3\frac{d^3}{ds^3}P_0(t, s, z)\right) \\
& -(c_2 - 3c_1)(T - t)\left(s\frac{d}{ds}P_0(t, s, z) + s^2\frac{d^2}{ds^2}P_0(t, s, z)\right) \\
& -(c_2 - 2c_1)(T - t)\left(-s\frac{d}{ds}P_0(t, s, z)\right),
\end{aligned}
\tag{30}
$$

where $c_1$ and $c_2$ are the same as those defined previously.

We summarize the above analysis and calculation on floating strike lookback put options in the following theorem.

**Theorem 3.** *Under the SV model governed by Equation* (1), *an approximation of the risk-neutral value P of a floating strike lookback put option is given by*

$$P = P_0 + \sqrt{\epsilon}P_1 + o(\sqrt{\varepsilon}), \tag{31}$$

*where $P_0$ and $P_1$ are given by Equations* (27) *and* (30), *respectively.*

## 6. Numerical Results and Sensitivity Analysis

In this section, we conduct a numerical study to investigate the sensitivity of the first-order correction term $P_1$ and our approximation results $P_0 + \sqrt{\epsilon}P_1$ with respect to the initial value of underlying asset. This means that we set $t = 0$ throughout this section. We also compare the results given by our closed form formulas with those generated by the Monte-Carlo simulation.

First of all, as conducted by Fouque et al. (2000, 2011) and Cao et al. (2021), we choose $f$ to take the following form:

$$f(y) = 0.35\left(\tan^{-1}(y) + \frac{\pi}{2}\right)/\pi + 0.05.$$

Secondly, the values of other parameters used in this section are given in Table 1 whenever they are required to be fixed.

**Table 1.** The role and numerical value of parameters.

| Parameter | Role | Value |
|:---:|:---:|:---:|
| $r$ | risk-free interest rate | 0.035 |
| $B$ | barrier level | 1500 |
| $K$ | put option strike price | 2700 |
| $T$ | maturity time | 1 |
| $c_1$ | as defined in Section 3 | $-0.004$ |
| $c_2$ | as defined in Section 3 | $-0.018$ |

Here, we do not choose precise values of $\beta$ and $\rho$, and particular forms of $\xi(y)$ (in Section 2) and $\phi(y)$ (in Section 3) to calculate the above values of $c_1$ and $c_2$. Instead, $c_1$ and $c_2$ are calibrated from the term structure of the implied volatility surface as described in the book of Fouque et al. (2000). Specifically, the implied volatility $I^\epsilon$ of a European vallina call option with fast mean-reverting stochastic process can be approximated by the following formula:

$$I^\epsilon = a\frac{\ln(\frac{K}{s})}{T-t} + b + o(\sqrt{\epsilon})$$

with

$$a = -\frac{c_1}{\langle f^2 \rangle^{3/2}} \quad \text{and} \quad b = \sqrt{\langle f^2 \rangle} + \frac{c_1}{\langle f^2 \rangle^{3/2}}\left(r + \frac{3}{2}\langle f^2 \rangle\right) - \frac{c_2}{\sqrt{\langle f^2 \rangle}}.$$

The parameters $a$ and $b$ are estimated as the slope and intercept of the regression fit of the observed implied volatilities as a linear function of logmoneyness-to-maturity-ratio $\ln(K/s)/(T-t)$. From the calibrated values $a$ and $b$ on the observed implied volatility surface, the parameters $c_1$ and $c_2$ are obtained as

$$c_1 = -a\sigma\langle f^2 \rangle^{3/2} \quad \text{and} \quad c_2 = \sqrt{\langle f^2 \rangle}((\sqrt{\langle f^2 \rangle} - b) - a(r + \frac{3}{2}\langle f^2 \rangle)).$$

Thirdly, note that when $t = 0$, $s = z$. Hence, in this case, the formula for $P_0$ given by Equation (27) is simplified.

Figure 1a shows how the $\sqrt{\epsilon}P_1$-term for a down-and-out barrier put option changes with respect to a variation in $\epsilon$ values As we can see, for fixed $\epsilon$, when $s$ increases, $P_1$ decreases first, and then increases after it hits its trough. When $\epsilon$ becomes smaller (equivalently, the mean-reverting speed becomes larger), $\sqrt{\epsilon}P_1$ approaches to a zero. Figure 1b shows how the value of $P_0 + \sqrt{\epsilon}P_1$ for a down-and-out put option varies with respect to the change in $\epsilon$ values. As we can see, when the value of $\epsilon$ changes from 0.01 to 0.0001, the value of $P_0 + \sqrt{\epsilon}P_1$ does not vary much. In fact, the values of $P_0 + \sqrt{\epsilon}P_1$ match well with the result of Monte-Carlo simulation in all cases. Furthermore, in all cases, the value of $P_0 + \sqrt{\epsilon}P_1$ declines as $s$ increases.

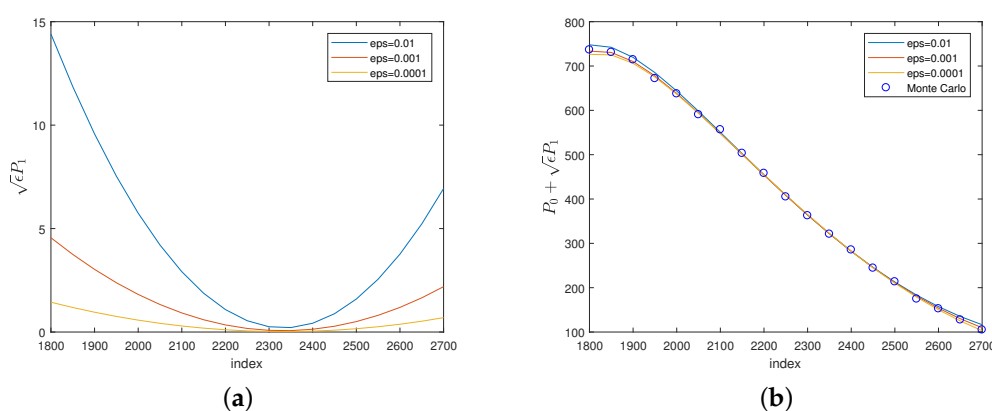

|  (a)  |  (b)  |

**Figure 1.** Plots of $\sqrt{\epsilon}P_1$ and $P_0 + \sqrt{\epsilon}P_1$ with different values of $\epsilon$ against the initial value of the underlying asset, for the down-and-out put option.

Figure 2a shows how the $\sqrt{\epsilon}P_1$-term for a floating strike lookback put changes with respect to a variation in $\epsilon$ values. In a similar pattern, for a fixed $\epsilon$-value, when $s$ increases, $P_1$ decreases first and then increases after it hits its trough. Similar to the case of down-and-out put options, when $\epsilon$ becomes smaller (equivalently, the mean-reverting speed becomes larger), $\sqrt{\epsilon}P_1$ approaches to zero. Figure 2b shows how the value of $P_0 + \sqrt{\epsilon}P_1$ for a floating strike put varies with respect to the change in $\epsilon$ values. When the value of $\epsilon$ changes from 0.01 to 0.001, the value of $P_0 + \sqrt{\epsilon}P_1$ varies. But when the value of $\epsilon$ changes from 0.001 to 0.0001, the value of $P_0 + \sqrt{\epsilon}P_1$ does not vary much. The values of $P_0 + \sqrt{\epsilon}P_1$ match well

with the result of Monte-Carlo simulation when $\epsilon = 0.001$ or $0.0001$. Furthermore, in all cases, the value of $P_0 + \sqrt{\epsilon}P_1$ increases as $s$ increases.

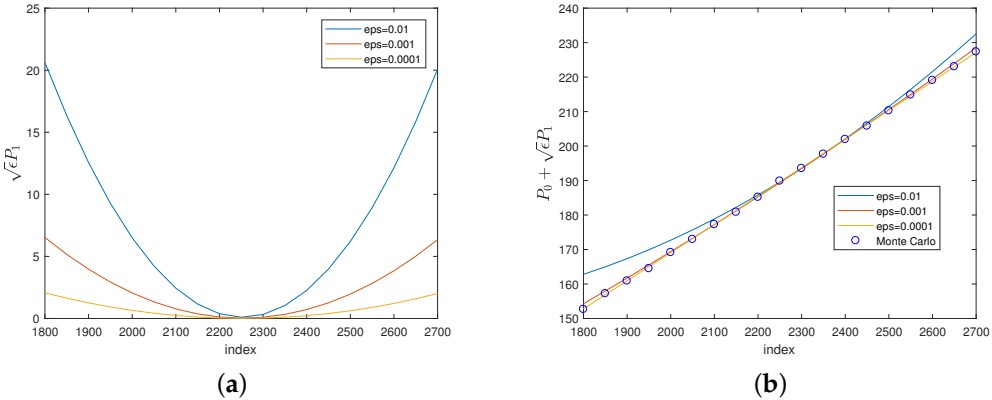

(a)　　　　　　　　　　　　　　　　　　(b)

**Figure 2.** Plots of $\sqrt{\epsilon}P_1$ and $P_0 + \sqrt{\epsilon}P_1$ with different values of $\epsilon$, against the initial value of the underlying asset, for floating strike put options.

Figure 3a illustrates the variation in the value of $P_0 + \sqrt{\epsilon}P_1$ for a down-and-out put option in response to changes in $\rho$ values. As $\rho$ shifts from $-0.6$ to $-0.4$, there is a slight decrease in the value of $P_0 + \sqrt{\epsilon}P_1$. Additionally, in all scenarios, the value of $P_0 + \sqrt{\epsilon}P_1$ shows an upward trend as $s$ increases.

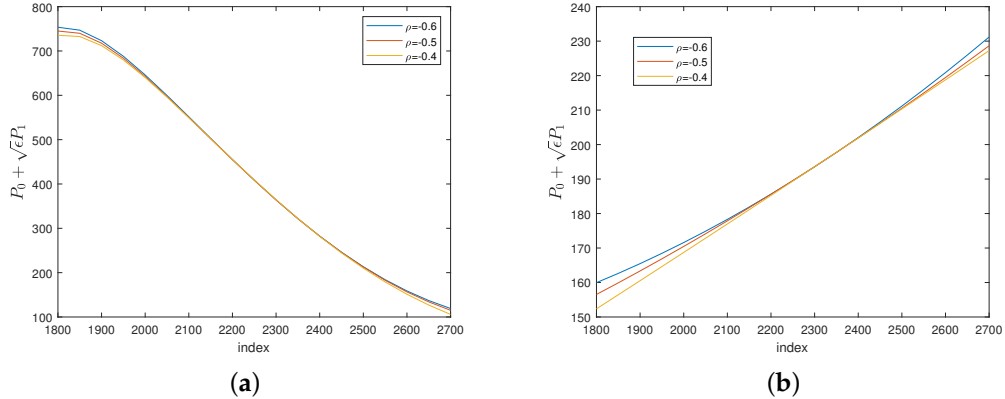

(a)　　　　　　　　　　　　　　　　　　(b)

**Figure 3.** Plots $P_0 + \sqrt{\epsilon}P_1$ with different values of $\rho$, against the initial value of the underlying asset, for down-and-out put option and floating strike put option.

Figure 3b depicts the change in the value of $P_0 + \sqrt{\epsilon}P_1$ for a floating strike put concerning variations in $\rho$ values. Similar to the previous case, a shift in $\rho$ from $-0.6$ to $-0.4$ results in a minor decline in the value of $P_0 + \sqrt{\epsilon}P_1$. Moreover, across all instances, an increase in $s$ is associated with a rise in the value of $P_0 + \sqrt{\epsilon}P_1$.

## 7. Concluding Remarks

This article establishes explicit closed-form solutions for first order approximations of down-and-out barrier and floating strike lookback put option prices under a stochastic volatility model by means of Mellin transform. The zero-order terms in the solutions for the prices of both types of put options coincide with those in Hull (2015) or Haug (2006) under the classical Back–Scholes model. Our numerical analysis shows that the results given by those explicit closed-form solutions match well with those generated by the Monte-Carlo simulation. This confirms the accuracy of the approximation. Furthermore, we also discussed the sensitivity of the first-order error terms and the approximation with respect to the underlying asset price and the mean-reverting speed of the OU-process which governs the volatility.

This model formula can be employed by financial professionals for the swift and precise pricing of barrier and lookback options. This is demonstrated by the efficiency of our formula in comparison to the conventional Monte-Carlo method. Our pricing formula offers an effective means of assessing barrier and lookback options. Looking ahead, we may extend our methodology to evaluate other path-dependent options in future works, including, but not limited to, Asian options, Russian options, and more.

**Author Contributions:** Conceptualization, J.C. ; methodology, J.C. and W.Z.; software, X.L.; validation, J.C. and W.Z.; formal analysis, X.L.; investigation, X.L.; resources, X.L.; data curation, X.L.; writing—original draft preparation, X.L.; writing—review and editing, J.C. and W.Z.; visualization, X.L.; supervision, J.C. and W.Z.; project administration, W.Z.; funding acquisition, J.C. All authors have read and agreed to the published version of the manuscript.

**Funding:** This research received no external funding.

**Data Availability Statement:** No new data were created or analyzed in this study. Data sharing is not applicable to this article.

**Acknowledgments:** The authors would like to thank Jeong-Hoon Kim for valuable discussions and suggestions on Mellin transform method. The authors also thank the anonymous reviewers for their careful reading of our manuscript and their many insightful comments and suggestions.

**Conflicts of Interest:** The authors declare no conflict of interest.

## Appendix A. Mellin Transform

The Mellin transform is an integral transform that may be regarded as the multiplicative version of the two-sided Laplace transform. It is often used in the theory of asymptotic expansions. For a locally Lebesgue integrable function $h : \mathbb{R}^+ \to \mathbb{R}$, the Mellin transform denoted by $\mathcal{M}h$ or $\hat{h}$ is given by

$$\hat{h}(w) = (\mathcal{M}h)(w) := \int_0^{+\infty} s^{w-1}h(s)\,ds, \quad w \in \mathbb{C},$$

and if $a < \mathrm{Re}(w) < b$ and $c$ such that $a < c < b$ exists, the inverse of the Mellin transform is expressed by

$$h(s) = \left(\mathcal{M}^{-1}\hat{h}\right)(s) = \frac{1}{2\pi i}\int_{c-i\infty}^{c+i\infty} s^{-w}\hat{h}(w)\,dw.$$

In this paper, we use the following properties of Mellin transform.

**Table A1.** List of properties of Mellin transform used in this paper.

| Function | Mellin Tansform |
|---|---|
| $h$ | $\hat{h}$ |
| $sh'$ | $-w\hat{h}$ |
| $s^2h''$ | $w(w+1)\hat{h}$ |
| $s^3h^{(3)}$ | $-w(w+1)(w+2)\hat{h}$ |
| $\frac{e^{\delta}s^{\eta}}{2\sqrt{\lambda\pi}}e^{-\frac{1}{4\lambda}(\ln s)^2}$ | $e^{\lambda(w+\eta)^2+\delta}$ |
| $sh' + s^2h''$ | $w^2\hat{h}$ |
| $-sh' - 3s^2h'' - s^3h^{(3)}$ | $w^3\hat{h}$ |

Here, $\lambda$, $\eta$, and $\delta$ are not related to $w$ or $s$, and $h'$, $h''$, and $h^{(3)}$ are the first-order, second-order, and third-order derivatives of $h$, respectively.

## Appendix B. Derivation of Formulas (20) and (27)

In this appendix, we give detailed derivation of the Formulas (20) and (27).

*Appendix B.1. Derivation of Formula* (20)

From Equation (19), we know that

$$
\begin{aligned}
P_0(t,s) \;=\; & \int_B^K (K-u)e^\delta \left(\frac{s}{u}\right)^\eta \left(\frac{1}{2\sqrt{\lambda\pi}} e^{-\frac{1}{4\lambda}\left(\ln\left(\frac{s}{u}\right)\right)^2}\right) \frac{du}{u} - \\
& \int_{\frac{B^2}{K}}^B \left(\frac{B}{u}\right)^{k_1-1}\left(K-\frac{B^2}{u}\right)e^\delta \left(\frac{s}{u}\right)^\eta \left(\frac{1}{2\sqrt{\lambda\pi}} e^{-\frac{1}{4\lambda}\left(\ln\left(\frac{s}{u}\right)\right)^2}\right)\frac{du}{u}.
\end{aligned}
$$

By letting $v = \ln u$, we convert the first integral to

$$
\begin{aligned}
& \int_{\ln B}^{\ln K} (K-e^v)s^\eta e^\delta e^{-\eta v}\left(\frac{1}{2\sqrt{\lambda\pi}} e^{-\frac{1}{4\lambda}(\ln s-v)^2}\right)dv \\
=\; & \frac{s^\eta e^\delta}{2\sqrt{\lambda\pi}}\left(\int_{\ln B}^{\ln K} Ke^{-\frac{1}{4\lambda}(v^2-2v\ln s+(\ln s)^2+4\lambda\eta v)}dv\right. \\
& \left. - \int_{\ln B}^{\ln K} e^{-\frac{1}{4\lambda}(v^2-2v\ln s+(\ln s)^2+4\lambda(\eta-1)v)}dv\right) \\
=\; & \frac{s^\eta e^\delta}{2\sqrt{\lambda\pi}}\left(\int_{\ln B}^{\ln K} Ke^{-\frac{1}{4\lambda}(v-\ln s+2\lambda\eta)^2+\lambda\eta^2-\eta\ln s}dv\right. \\
& \left. - \int_{\ln B}^{\ln K} e^{-\frac{1}{4\lambda}[v-\ln s+2\lambda(\eta-1)]^2+\lambda(\eta-1)^2-(\eta-1)\ln s}dv\right),
\end{aligned}
$$

we further apply the following changes in variables:

$$
x' := \frac{v-\ln s+2\lambda\eta}{\sqrt{2\lambda}} \quad\text{and}\quad x'' := \frac{v-\ln s+2\lambda(\eta-1)}{\sqrt{2\lambda}}
$$

to obtain

$$
\begin{aligned}
& \int_{\ln B}^{\ln K} (K-e^v)s^\eta e^\delta e^{-\eta v}\left(\frac{1}{2\sqrt{\lambda\pi}} e^{-\frac{1}{4\lambda}(\ln s-v)^2}\right)dv \\
=\; & \frac{e^\delta}{\sqrt{2\pi}}\left(Ke^{\lambda\eta^2}\int_{\frac{\ln(\frac{B}{s})+2\lambda\eta}{\sqrt{2\lambda}}}^{\frac{\ln(\frac{K}{s})+2\lambda\eta}{\sqrt{2\lambda}}} e^{-\frac{x'^2}{2}}dx' - se^{\lambda(\eta-1)^2}\int_{\frac{\ln(\frac{B}{s})+2\lambda(\eta-1)}{\sqrt{2\lambda}}}^{\frac{\ln(\frac{K}{s})+2\lambda(\eta-1)}{\sqrt{2\lambda}}} e^{-\frac{x''^2}{2}}dx''\right) \\
=\; & Ke^{\delta+\lambda\eta^2}\left[\Phi\left(\frac{\ln(\frac{K}{s})+2\lambda\eta}{\sqrt{2\lambda}}\right) - \Phi\left(\frac{\ln(\frac{B}{s})+2\lambda\eta}{\sqrt{2\lambda}}\right)\right] \\
& -se^{\delta+\lambda(\eta-1)^2}\left[\Phi\left(\frac{\ln(\frac{K}{s})+2\lambda(\eta-1)}{\sqrt{2\lambda}}\right) - \Phi\left(\frac{\ln(\frac{B}{s})+2\lambda(\eta-1)}{\sqrt{2\lambda}}\right)\right].
\end{aligned}
$$

Now, if we plug into $\delta$, $\eta$, and $\lambda$ into the above formula, we derive

$$
\begin{aligned}
& \int_{\ln B}^{\ln K} (K-e^v)s^\eta e^\delta e^{-\eta v}\left(\frac{1}{2\sqrt{\lambda\pi}} e^{-\frac{1}{4\lambda}(\ln s-v)^2}\right)dv \\
&= Ke^{-r(T-t)}\left[\Phi\left(-\Delta_-\left(\frac{s}{K}\right)\right) - \Phi\left(-\Delta_-\left(\frac{s}{B}\right)\right)\right] \\
& -s\left[\Phi\left(-\Delta_+\left(\frac{s}{K}\right)\right) - \Phi\left(-\Delta_+\left(\frac{s}{B}\right)\right)\right].
\end{aligned}
$$

Similarly, we can evaluate the second integral

$$
\int_{\frac{B^2}{K}}^B \left(\frac{B}{u}\right)^{k_1-1}\left(K-\frac{B^2}{u}\right)e^\delta \left(\frac{s}{u}\right)^\eta \left(\frac{1}{2\sqrt{\lambda\pi}} e^{-\frac{1}{4\lambda}\left(\ln\left(\frac{s}{u}\right)\right)^2}\right)\frac{du}{u}
$$

to obtain

$$Ke^{-r(T-t)}\left(\frac{B}{s}\right)^{k_1-1}\left[\Phi\left(\Delta_-\left(\frac{B}{s}\right)\right)-\Phi\left(\Delta_-\left(\frac{B^2}{sK}\right)\right)\right]$$
$$-B\left(\frac{B}{s}\right)^{k_1}\left[\Phi\left(\Delta_+\left(\frac{B}{s}\right)\right)-\Phi\left(\Delta_+\left(\frac{B^2}{sK}\right)\right)\right].$$

Putting these two integrals together yields Formula (20).

*Appendix B.2. Derivation of Formulas* (27)

From Equation (26), we have

$$Q_0(t,u) = \int_0^1 (1-\xi)e^\delta\left(\frac{u}{\xi}\right)^\eta\left(\frac{1}{2\sqrt{\lambda\pi}}e^{-\frac{1}{4\lambda}\left(\ln\left(\frac{u}{\xi}\right)\right)^2}\right)\frac{d\xi}{\xi}+$$

$$\int_1^\infty\left(-\frac{1}{k_1}\xi^{1-k_1}+\frac{\xi}{k_1}\right)e^\delta\left(\frac{u}{\xi}\right)^\eta\left(\frac{1}{2\sqrt{\lambda\pi}}e^{-\frac{1}{4\lambda}\left(\ln\left(\frac{u}{\xi}\right)\right)^2}\right)\frac{d\xi}{\xi}.$$

We let $v=\ln\xi$. For the first integral, we have

$$\int_0^1(1-\xi)e^\delta\left(\frac{u}{\xi}\right)^\eta\left(\frac{1}{2\sqrt{\lambda\pi}}e^{-\frac{1}{4\lambda}\left(\ln\left(\frac{u}{\xi}\right)\right)^2}\right)\frac{du}{u}$$

$$=\int_{-\infty}^0 u^\eta(1-e^v)e^{\delta-v\eta}\left(\frac{1}{2\sqrt{\lambda\pi}}e^{-\frac{1}{4\lambda}(\ln u-v)^2}\right)dv$$

$$=\frac{u^\eta e^\delta}{2\sqrt{\lambda\pi}}\left(\int_{-\infty}^0 e^{-\frac{1}{4\lambda}\left(v^2-2v\ln u+(\ln u)^2+4\lambda\eta v\right)}dv\right.$$
$$\left.-\int_{-\infty}^0 e^{-\frac{1}{4\lambda}\left(v^2-2v\ln u+(\ln u)^2+4\lambda(\eta-1)v\right)}dv\right)$$

$$=\frac{u^\eta e^\delta}{2\sqrt{\lambda\pi}}\left(\int_{-\infty}^0 e^{-\frac{1}{4\lambda}(v-\ln u+2\lambda\eta)^2+\lambda\eta^2-\eta\ln u}dv\right.$$
$$\left.-\int_{-\infty}^0 e^{-\frac{1}{4\lambda}(v-\ln u+2\lambda(\eta-1))^2+\lambda(\eta-1)^2-(\eta-1)\ln u}dv\right).$$

Next, we let

$$v':=\frac{v-\ln u+2\lambda\eta}{\sqrt{2\lambda}}\quad\text{and}\quad v'':=\frac{v-\ln u+2\lambda(\eta-1)}{\sqrt{2\lambda}}.$$

Then, we have

$$\int_0^1(1-\xi)e^\delta\left(\frac{u}{\xi}\right)^\eta\left(\frac{1}{2\sqrt{\lambda\pi}}e^{-\frac{1}{4\lambda}\left(\ln\left(\frac{u}{\xi}\right)\right)^2}\right)\frac{du}{u}$$

$$=\frac{e^\delta}{\sqrt{2\pi}}\left(\int_{-\infty}^{\frac{-\ln u+2\lambda\eta}{\sqrt{2\lambda}}}e^{-\frac{v'^2}{2}+\lambda\eta^2}dv'-u\int_{-\infty}^{\frac{-\ln u+2\lambda(\eta-1)}{\sqrt{2\lambda}}}e^{-\frac{v''^2}{2}+\lambda(\eta-1)^2}dv''\right)$$

$$=e^{\delta+\lambda\eta^2}\Phi\left(\frac{-\ln u+2\lambda\eta}{\sqrt{2\lambda}}\right)-ue^{\delta+\lambda(\eta-1)^2}\Phi\left(\frac{-\ln u+2\lambda(\eta-1)}{\sqrt{2\lambda}}\right)$$

$$=e^{-r(T-t)}\Phi\left(-\Delta_-\left(\frac{s}{z}\right)\right)-\left(\frac{s}{z}\right)\Phi\left(-\Delta_+\left(\frac{s}{z}\right)\right).$$

For the second integral, we have

$$\int_1^\infty \left( -\frac{1}{k_1}\xi^{1-k_1} + \frac{\xi}{k_1} \right) e^\delta \left( \frac{u}{\xi} \right)^\eta \left( \frac{1}{2\sqrt{\lambda\pi}} e^{-\frac{1}{4\lambda}\left(\ln\left(\frac{u}{\xi}\right)\right)^2} \right) \frac{d\xi}{\xi}$$

$$= \int_0^\infty \left( -\frac{1}{k_1}e^{(1-k_1)v} + \frac{1}{k_1}e^v \right) e^\delta u^\eta e^{-v\eta} \left( \frac{1}{2\sqrt{\lambda\pi}} e^{-\frac{1}{4\lambda}(\ln u - v)^2} \right) dv$$

$$= \frac{e^\delta u^\eta}{2k_1\sqrt{\lambda\pi}} \int_0^\infty \left( -e^{\eta v - \frac{1}{4\lambda}(\ln u - v)^2} + e^{v(1-\eta) - \frac{1}{4\lambda}(\ln u - v)^2} \right) dv$$

$$= \frac{e^\delta u^\eta}{2k_1\sqrt{\lambda\pi}} \left( \int_0^\infty -e^{-\frac{1}{4\lambda}(v - \ln u - 2\lambda\eta)^2 + \lambda\eta^2 + \eta \ln u} dv \right.$$

$$\left. + \int_0^\infty e^{-\frac{1}{4\lambda}(v - \ln u - 2\lambda(1-\eta))^2 + \lambda(1-\eta)^2 + (1-\eta)\ln u} dv \right),$$

where we use the fact that $k_1 - 1 + \eta = -\eta$. Furthermore, we introduce a new variable

$$v''' := \frac{v - \ln u - 2\lambda\eta}{\sqrt{2\lambda}}.$$

Then, we have

$$\int_1^\infty \left( -\frac{1}{k_1}\xi^{1-k_1} + \frac{\xi}{k_1} \right) e^\delta \left( \frac{u}{\xi} \right)^\eta \left( \frac{1}{2\sqrt{\lambda\pi}} e^{-\frac{1}{4\lambda}\left(\ln\left(\frac{u}{\xi}\right)\right)^2} \right) \frac{d\xi}{\xi}$$

$$= \frac{e^\delta u^\eta}{k_1\sqrt{2\pi}} \left( \int_{\frac{-\ln u - 2\lambda\eta}{\sqrt{2\lambda}}}^\infty -e^{-\frac{v'''^2}{2}} e^{\lambda\eta^2 + \eta \ln u} dv''' \right.$$

$$\left. + \int_{\frac{-\ln u + 2\lambda(\eta-1)}{\sqrt{2\lambda}}}^\infty e^{-\frac{v''^2}{2}} e^{\lambda(\eta-1)^2 + (1-\eta)\ln u} dv'' \right)$$

$$= -\frac{1}{k_1}e^{\delta + \lambda\eta^2} u^{1-k_1} \Phi\left( \frac{\ln u + 2\lambda\eta}{\sqrt{2\lambda}} \right) + \frac{1}{k_1}u e^{\delta + \lambda(\eta-1)^2} \Phi\left( \frac{\ln u + 2\lambda(1-\eta)}{\sqrt{2\lambda}} \right)$$

$$= -\frac{1}{k_1}\left( \frac{s}{z} \right)^{1-k_1} e^{-r(T-t)} \Phi\left( -\Delta_-\left( \frac{z}{s} \right) \right) + \frac{1}{k_1}\left( \frac{s}{z} \right) \Phi\left( \Delta_+\left( \frac{s}{z} \right) \right).$$

Putting these two integrals together and using the fact that $P_0 = zQ_0$, we can obtain our Formula (27).

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
