# Peer review of "Pricing Path-Dependent Options under Stochastic Volatility via Mellin Transform"

_jrfm, doi:10.3390/jrfm16100456_

Round 1

Reviewer 1 Report

The English language needs little editing.

Author Response

We thank the anonymous reviewer for the careful reading of our manuscript and many insightful comments and suggestions. Attached are our responses for each comment.

Reviewer 2 Report

Dear Authors, 

The intricacies of option pricing, especially concerning path-dependent options under stochastic volatility, have taken centre stage in financial mathematics and quantitative finance. With financial markets becoming increasingly complex and unpredictable, the quest for precise, reliable, and efficient pricing models has never been more vital. The application of mathematical tools, such as the Mellin transform, presented in the paper, reflects the ongoing evolution and sophistication of methodologies addressing these challenges.

The authors provided highly formalised initial results. However, there are some minor recommendations for a rise in the value of the results may be formulated:

1. In the "Numerical Results and Sensitivity Analysis" section, the authors may clarify the statistical rationale for choosing the Monte Carlo model parameters. For example, like the approach applied in previous work: Cao, J., J.-H. Kim, and W. Zhang. 2021. "Pricing variance swaps under hybrid CEV and stochastic volatility." Journal of Computational and Applied Mathematics Article.

2. The conclusion section is compact; however, it would be beneficial for the authors to elucidate the novelty of their findings distinctly. Additionally, emphasizing the practical significance for investors would enhance the paper's applicability.

Overall, the paper is well-structured, has a rigorous scientific tone, and can be recommended for publication after some minor revisions.

Author Response

(The authors gave the same response as above.)

Reviewer 3 Report

The purpose of the research is to derive closed-form formulas of first-order approximation for down-and- out barrier and floating strike lookback put option prices under a stochastic volatility model, by using an asymptotic approach. It looks like an impressive mathematical demonstration. Please note that mathematics is not my specialization.

From my point of view, I would recommend to work more on practical utility of your research. Why should we read your paper? What can we do with your findings? Who can use these findings and how? Do your paper advance knowledge in the field?

 I would also recommend to update the list of references with more recently published papers. The list seems to be old.

Author Response

(The authors gave the same response as above.)
